# Can Computational Intelligence Model Phenomenal Consciousness?

Eduardo C. Garrido Merchán [1,*,†] and Sara Lumbreras [2,†]

1 Department of Quantitative Methods, Universidad Pontificia de Comillas, 28049 Madrid, Spain
2 Institute for Research in Technology IIT, Universidad Pontificia Comillas, 28015 Madrid, Spain; slumbreras@comillas.edu
* Correspondence: ecgarrido@icade.comillas.edu
† These authors contributed equally to this work.

**Abstract:** Consciousness and intelligence are properties that can be misunderstood as necessarily dependent. The term artificial intelligence and the kind of problems it managed to solve in recent years has been shown as an argument to establish that machines experience some sort of consciousness. Following Russell's analogy, if a machine can do what a conscious human being does, the likelihood that the machine is conscious increases. However, the social implications of this analogy are catastrophic. Concretely, if rights are given to entities that can solve the kind of problems that a neurotypical person can, does the machine have potentially more rights than a person that has a disability? For example, the autistic syndrome disorder spectrum can make a person unable to solve the kind of problems that a machine solves. We believe the obvious answer is no, as problem-solving does not imply consciousness. Consequently, we will argue in this paper how phenomenal consciousness, at least, cannot be modeled by computational intelligence and why machines do not possess phenomenal consciousness, although they can potentially develop a higher computational intelligence than human beings. In order to do so, we try to formulate an objective measure of computational intelligence and study how it presents in human beings, animals, and machines. Analogously, we study phenomenal consciousness as a dichotomous variable and how it is distributed in humans, animals, and machines.

**Keywords:** computational intelligence; phenomenal consciousness





## 1. Introduction

The concept of consciousness has remained difficult to define and understand [1]. Consciousness (at least, phenomenal consciousness, which will be our main focus) is an ontologically subjective phenomenon, and the scientific method applies only to epistemologically objective phenomena [2]. Although remarkable progress has been made concerning the definition of physical correlates of consciousness [3], the subjective experience of consciousness remains elusive. What is more, the multiple realizability assumption means that there might be different paths to consciousness, with different material substrates or physical correlates, so the evaluation of correlates is not equivalent to the identification of consciousness.

We can split the consciousness concept into several associated phenomena. Ned Block [4] argued that discussions on consciousness often fail to adequately differentiate between two distinct aspects: phenomenal consciousness (P-consciousness) and access consciousness (A-consciousness). It should be noted that these concepts predate Block. Phenomenal consciousness refers to raw experiences, such as moving, colored forms, sounds, sensations, emotions, and feelings, with our bodies and responses at the core. These experiences, detached from their impact on behavior, are referred to as qualia. On the other hand, A-consciousness pertains to the accessibility of information in our minds for verbal reports, reasoning, and behavioral control. Perception provides access-conscious information about what we perceive, introspection grants access to information about

our thoughts, and remembrance provides access-conscious information about the past, and so forth. Attention is the mechanism that focuses consciousness on a particular set of information coming from perception, interoception, or remembrance. Although some philosophers, like Daniel Dennett, [5] question the validity of this distinction, others generally accept it.

While Block's classification of two types of consciousness has been influential, some philosophers, like William Lycan [6], propose a more extensive range. Lycan identifies at least eight distinct types of consciousness, including organism consciousness, control consciousness, the consciousness of, state/event consciousness, reportability, introspective consciousness, subjective consciousness, and self-consciousness. However, even this list excludes other less recognized forms. Arguably, the most recognizable and relevant forms of consciousness are phenomenal consciousness, access consciousness, and self-consciousness, which are defined as the models of our identity that we build based on our experience [7].

Debate persists regarding the coexistence or separability of the different forms of consciousness, especially access consciousness and phenomenal consciousness, and how differently to approach each of them, which, as explained, would be identified with the hard and soft problems, respectively.

The deep divide lies, in other words, between the subjective phenomenon, the relative to the observer experience of something that the observer is paying attention to, and the objective phenomena, which can be simulated in a computer even if it is related to consciousness [8]. For example, we can manipulate colors with computers but we cannot simulate qualia. We can even simulate the "bottleneck of consciousness" [9] but cannot simulate the perceived experience of the result.

There is already a vast array of literature that deals with machine simulation of particular aspects of consciousness (see [10] for a well-structured and comprehensive review). The main approaches have been the global workspace model, information integration, an internal self-model, a higher-level representation, and attention mechanisms.

Global workspace theory [11] views the brain as a collection of specialized processors that provide for sensation, motor control, language, reasoning, and so forth. Conscious experience is hypothesized to emerge from globally shared information.

Information integration explains consciousness as shared or mutual information shared among brain regions as they interact in a constructive manner [12].

The internal self-model is based on the idea that our mind includes a model of our body and how it relates to the space it is embedded in that is supported by certain brain regions [13].

The higher-level representation theory explains consciousness with a higher pattern of information encoding, akin to symbolic processing [14].

Finally, some authors point to the attention mechanisms that filter the fraction of ongoing experience that is experienced as the basis of consciousness. This is one of the mechanisms that has been simulated more successfully, even being considered at the core of the recent revolution of generative AI [15]. Another relevant example has been the simulation of a computational ontology of a person based on historical records [16].

All the previous models have lent themselves to simulation, which has been a fertile terrain for understanding some of the specific mechanisms of consciousness. These exercises have even resulted in the description of interesting related phenomena, such as in Axiomatic Consciousness Theory, a specialized simulation of visual processing based on attention, which can explain some properties of visual experiences, such as foveal, eye field, in-front, and the space images [17].

However, how to implement or simulate phenomenal consciousness remains a complete mystery. We cannot simulate the experience of the perception of qualia, such as the redness of the red color.

We recall the example in the "Mary's mind experiment", a blind girl that is an expert in vision. Although she is an expert, Mary is unable to know what is the difference in the perception of red or black, as this information comes from qualia and is subjective,

that is, cannot be represented in an objective manner such as in a book description [18]. This is known as the knowledge argument: and it rests on the idea that someone who has completely objective knowledge, from the point of view of the scientific method and our epistemological scope, about another conscious being might yet lack knowledge about how it feels to have the experiences of that being [19].

A common belief shared by part of the computer science community, and concretely in the machine consciousness community [20] and even in part of the philosophy of mind community and more concretely shared by the connectionism community [21], is that if an artificial general intelligence [22] is modeled (supposedly via some meta-learning [23] or transfer learning methodology [24] applied to high capacity deep learning models [25] and huge datasets), due to emergence, phenomenal consciousness may arise. Hence, this group of people believes, assuming the multiple realizability philosophy of mind assumption [26], that an intelligent enough system is the cause of phenomenal consciousness. Phenomenal consciousness arises then as an epiphenomenon [27], or, alternatively, intelligence is the cause of phenomenal consciousness or vice-versa. To justify this relation, it is mandatory to provide an objective definition of intelligence, which is problematic as the different mentioned communities provide different definitions of intelligence, particularly computational intelligence. In order to clarify our definition of intelligence, we provide a clarification attempt in Section 4.

Several objections have been made to Mary's argument; for example, that qualia are not information. Hence, Mary would know everything about color. However, we argue that qualia are indeed phenomenological information (of the form "How it feels to"), being modeled in the qualia space $Q$ designed by the neuroscientific information integration theory of Tononi [12]. In particular, Q has an axis for each possible state (activity pattern) of an information complex (please see Tononi's paper for more details [12]). Within Q, each sub mechanism specifies a point corresponding to a repertoire of system states. Most critically, arrows between repertoires in Q define informational relationships. Consequently, and given the mathematical information theory [28], qualia is information that would be unattainable for Mary if she is blind.

In order to continue analyzing the potential statistical, or even metaphysical, causal relation between phenomenal consciousness and intelligence, it is important to also briefly describe intelligence. Coming from the psychology community, and in a broad sense, intelligence is a very general mental capability that, among other things, involves the ability to reason, plan, solve problems, think abstractly, comprehend complex ideas, learn quickly, and learn from experience [29]. If these problems are computational, we can reduce and quantify intelligence as an analytical expression [30], giving rise to the concept of computational intelligence. We can express it quantitatively and study its relation with phenomenal consciousness. Here, we argue that computational intelligence would be an ontologically objective continuous numerical latent variable whose observation is noisy and obscured by a series of factors such as the personality or mood of the person being measured.

Computational intelligence can be exhibited by either living beings or machines, giving rise to what we call machine intelligence [31]. However, AI does not involve phenomena such as understanding [32], as understanding requires an entity to be aware of the learned concept. Nevertheless, computational intelligence does not require understanding. For example, a model can beat any human at chess, while being unaware of doing so. Hence, we find that computational intelligence, which is a subset of intelligence, does not share a relation in this example with phenomenal consciousness, which is the focus of this paper and we will further continue to provide examples such as this one.

The organization of this paper is as follows. First, we illustrate the analogy of Russell, which is at the root of the belief that intelligence and consciousness are related, and formalize it from a Bayesian point of view. Then, we provide some simple counter-examples that show empirical evidence of how unlikely it is for the hypothesis to be true. In an additional section, we study the concept of intelligence and provide a new definition of

computational intelligence to more formally reject the mentioned hypothesis. With that definition, we study the potential causal relationship between phenomenal consciousness and computational intelligence. Afterward, we formalize how computational intelligence is not able to model phenomenal consciousness. Finally, we finish the paper with a section of conclusions and further work.

## 2. The Social Risks of a Consciousness Directly Correlated with Intelligence

Making the assumption that intelligence and consciousness are related can lead to catastrophic consequences for society. For instance, this assumption can lead to the unjust treatment of individuals with low intelligence, such as those with Down syndrome or severe forms of autism. For example, according to an IQ test, an autistic person may seem much less intelligent than they are [33]. If they are believed to be less conscious, they could be granted lesser levels of assistance. This assumption can result in neglect and discrimination towards individuals with low intelligence. Concretely, one example of a government treating individuals with cognitive disabilities unfairly is the case of the eugenics movement in the United States. The movement, which gained traction in the early 20th century, advocated for the forced sterilization of individuals deemed to have "feeble-mindedness" or other cognitive disabilities [34]. This was based on the belief that such individuals were a burden to society and could not contribute to it in any meaningful way. Many individuals with cognitive disabilities were sterilized against their will, and this practice continued until the mid-20th century [35]. Such policies could be repeated if the belief that individuals with cognitive disabilities or low intelligence were seen as less conscious, and therefore did not require assistance or equal treatment. Therefore, it is essential to recognize that intelligence and consciousness are complex and multifaceted concepts that should not be reduced to a single measure or assumption.

## 3. Russell's Analogy of Consciousness

In this section, we will present the analogy postulated by Russell about intelligence and consciousness [36]. Broadly speaking, he states that it is highly probable that consciousness is the only cause of the intelligent behavior that humans exhibit. It does so by supposing that if the behavior of people is similar to our own, then, by observation, we can establish a causal relation that the other people possess consciousness as we do. Literally, from the Analogy of Russell, we have that:

> "We are convinced that other people have thoughts and feelings that are qualitatively fairly similar to our own...it is clear that we must appeal to something that may be vaguely called analogy. The behavior of other people is in many ways analogous to our own, and we suppose that it must have analogous causes. What people say is what we should say if we had certain thoughts, and so we infer that they probably have these thoughts...As it is clear to me that the causal laws governing my behavior have to do with thoughts...how do you know that the gramophone does not think?...it is probably impossible to refute materialism by external observation alone. If we are to believe that there are thoughts and feelings other than our own, that must be in virtue of some inference in which our own thoughts and feelings are relevan t...establish a rational connection between belief and data...From subjective observation I know that A, which is a thought or feeling, causes B, which is a bodily act, whatever B is an act of my own body, A is its cause. I now observe an act of the kind B in a body not my own, and I am having no thought or feeling of the kind A. But I still believe on the basis of self-observation, that only A can cause B. I, therefore, infer that there was an A which caused B, though it was not an A that I could observe."

Russell's analogy could be roughly summarized as "consciousness, which we cannot observe can be inferred by behavior, which we can observe". Russell refers to the causal laws going from thoughts to behavior. If we understand "having thoughts" in the hard sense, as the subjective experience in phenomenal consciousness, then for Russell consciousness would be the cause of behavior (in particular, of intelligent behavior).

However, we must realize that this reasoning falls prey to the fallacy of affirmation of the consequent: there are many reasons why an agent may exhibit intelligent behavior. For instance, the DALLE-2 model generates artistic images but almost all of us would agree it is not conscious. The same can be said of ChatGPT and its dialogue applications, even in a stronger manner. The analogy reasoning involves more correlation than causality. Here, the confounder would be that both human behavior and the behavior of Generative AI are both the result of human intent.

From a classical logic point of view, Russell states that every living being produces intelligent behavior, applying modus ponens. However, applying modus tollens if a being does not exhibit intelligent behavior outside, then it would not be conscious, at least, from the probabilistic point of view that is considered in the analogy. However, this reasoning is flawed. For instance, a person suffering from severe autism, may not show intelligent behavior; hence, following the argument of Russell, there is, at least, a high likelihood that this person is not conscious. However, this is not true.

Having shown that there are phenomenally conscious human beings that do not exhibit intelligent behavior according to several estimates or that their computational intelligence cannot be compared to the ones of computers, we provide another counter-example to the analogy, coming from the field of artificial intelligence [37]. In particular, we have seen how, in recent years, due to methodological advantages such as deep learning [25] and the rise of computational power, intelligent systems have surpassed human abilities in a series of complex games. Some examples include AlphaGo winning at the Go game to the world champion [38], IBM Watson winning at Jeopardy [39], and discovering new unknown chess strategies with deep learning [40]. General intelligence is a broad property, in the ontological sense, but we can reduce its meaning and provide a definition for a subset of it. In particular, we can epistemologically measure it as a function of the proportion of the computational problems that a system can solve from the set of all computational problems. Following this lower bound of general intelligence, a system implementing all the known machine learning models and meta-models of them able to solve any task with sufficient data will, for sure, outperform the performance of human beings in a broad series of problems and even solve problems that we do not know how to solve, like the protein folding problem [41].

Following the analogy of Russell, it would seem highly likely that a system that implements these algorithms would be conscious. It would be even more probable that such a system is more conscious than any other human being. However, as we will further show, providing multi-disciplinary arguments, the likelihood that a Turing machine, which is essentially any known software being executed by a computer, is conscious is almost zero. Hence, the evidence given by the data shows that the hypothesis that there is a causal relation between intelligence and consciousness is fallacious. Finally, as an extreme argument, we can measure the computational intelligence, as in the next section we will do, of a severely autistic person with respect to a system implementing several methodologies such as AlphaGo, showing that phenomenal consciousness cannot be modeled by computational intelligence.

## 4. Defining Intelligence

Intelligence is a widely known concept that has been assimilated by the computer-science community to coin the term artificial intelligence. However, artificial intelligence is a misleading term, as it requires a proper definition of intelligence as a property that can be modeled with a set of numerical variables.

In particular, multiple definitions of intelligence have been proposed by different communities but all of them seem to be a reduction of the general meaning of intelligence. For example, if we include as intelligence the ability to understand and empathize with another person, this ability requires us to feel the situation that is having the other person. From a theoretical, relative to the observer and internal point of view, it will not be enough

to appear to understand or feel by simulation methods based on quantitative measures, it would need to receive the qualia of the feeling or the idea being understood.

Hence, feeling requires awareness, or phenomenal consciousness, of the person that is having a conversation. As a consequence, this ability cannot be reduced to a simple set of numerical variables nor be implemented in a machine. In this section, we make a review of some of the different definitions of intelligence, to further justify why computational intelligence cannot model phenomenal consciousness.

### 4.1. Artificial Intelligence and Deep Learning

Artificial intelligence [37] has another controversial definition. Generally, it is the science and engineering of making intelligent machines [42]. But, if we want to define the intelligence of machines, that leads to a circular definition. We prefer to define it as an objective quantitative measure that is determined by the scope of problems that an artificial system is able to solve.

In recent years, due to the significant advances in computational power, it has been possible to implement high-capacity machine-learning models [43] like deep neural networks, which is usually referred to as deep learning [25]. As we have illustrated in the introduction, these models, whose capacity includes having more than 500 billion of parameters [44], are able to solve complex problems like the protein folding problem [41], go and chess [38], write philosophy articles in a newspaper mocking the type of writings that were usually only attributed to human beings [45] mastering natural language processing and common sense tasks and generating art [46]. In essence, deep learning methodologies are able to fit complex probability distributions by being able to generalize their behavior to tasks that are only supposed to be solved by humans, making their behavior indistinguishable from that of humans [47].

However, deep neural networks are software programs that are executed using computer hardware in a CPU (Central Processing Unit), GPU (Graphical Processing Unit), or TPU (Tensorial Processing Unit). Concretely, these hardware units are part of a Von Neumann architecture, which is essentially a Turing machine, making deep neural network algorithms that can be executed by a Von Neumann architecture, hence a Turing machine. Consequently, as we will further provide arguments for this claim, they lack awareness or phenomenal consciousness. As a result, they are unable to understand nor experience the scope of problems that they are solving and merely solve computations involving pattern recognition, independently of their complexity. Hence, artificial intelligence systems (at least in their current form) only possess computational intelligence [30,31], lacking understanding as it requires the qualia of the problem being solved. However, a virtue of computational intelligence is that it can be quantified, as it solves objective problems belonging to the set of all possible computational problems. In contrast, general human intelligence, as we will further see, is subjective and relative to the observer, requiring the qualia generated by understanding, feelings, or empathy and hence impossible to quantify without reducing it at its essence.

There are several propositions to quantify the computational intelligence that a system or an entity possesses. Let $\pi$ be an entity, for example, a human being, that in every instant $t$ is able to perform a set of actions $A$ to solve a given problem. An intelligent agent $\pi$ would decide, for every instant $t$, the optimum action $a^\star \in A$ to solve the problem. The branch of computer science that studies how to train intelligent agents in this framework is called reinforcement learning [48] and can be directly extrapolated to reality. For example, if we want to say the optimum phrase to win a negotiation, in every instant $t$ we receive the sentence of the person that we are negotiating with, her word frequency distributions, or her emotional state. As a function of all that information, we choose to answer a certain phrase in a particular manner. As we can see, reinforcement learning can be applied to a plethora of computational intelligence problems. In fact, reinforcement learning systems are implemented in robots for planning. We now introduce the analytical expressions of several measures of intelligence to objectively clarify how they could be modeled mathematically.

Dealing with these systems, which can perfectly be humans, the universal intelligence function Y of a data structure resembling an agent $\pi$ is given by the following measure [49]:

$$Y(\pi) = \sum_{\mu \in E} 2^{-K(\mu)} V_\mu^\pi. \tag{1}$$

where $\mu$ is a data structure representing an environment from the set $E$ of all computable reward bounded environments, $K(\cdot)$ is the Kolmogorov complexity, and $V_\mu^\pi := \mathbb{E}(\sum_{i=1}^\infty R_i)$ is the expected sum of future rewards $R_i$ when agent $\pi$ interacts with environment $\mu$. That is, the previous expression is a weighted average of how many problems $\mu \in E$ an agent $\pi$ can solve weighted by their difficulty $2^{-K(\mu)} V_\mu^\pi$. In particular, this is the reason why $V_\mu^\pi$ includes $\pi$. Several things are interesting in this expression. First, the set of all computable reward bounded environments, i.e., would be the set of all computational problems and is countably infinite. Hence, the intelligence Y of an agent is not upper-bounded. If we transform the set $E$ to a set where the area of a problem $\mu$ is given as a function of its difficulty $S(\mu)$, with a larger area given to more difficulty with respect to a particular agent $\pi$, the previous measure can be transformed in this abstract, general measure:

$$Y(\pi) = \int_E \delta(\mu \mid \pi) S(\mu) d\mu. \tag{2}$$

where $S(\mu)$ is an oracle function that gives the objective area of a problem $\mu$ and $\delta(\mu \mid \pi)$ is a delta function representing whether the particular problem is solved or not by the agent $\pi$. Recall that the delta function outputs 1 if the problem is solved and 0 otherwise. As the set is potentially countably infinite, a problem can be decomposed according to the progress on it in different problems until a simple base problem, each one with a different area to measure the progress of an agent in the progress of a particular problem. Interestingly, the integral over the set $E$ gives the area of computational problems being solved, and this area is infinite. Moreover, an oracle giving a particular objective unbiased measure of difficulty for every problem would be needed. Depending on the features of the system, a problem may be more difficult than others, especially for non-computable problems requiring qualia to be solved. These objections make such a measure impossible to be unbiasedly implemented in practice but may be a lower bound of the computational intelligence of a system, animal, or human being.

Another example of an intelligence measure of a system represented by *IS* for a scope of tasks sampled from $P_{scope}$ is now described. We have generalized from the measure proposed by Chollet, taking into account not only the scope of particular tasks that are numerable into a set but all the possible tasks that can be performed in our universe, which is potentially infinite and the one that we believe should be taken into account. Recall that we want to provide an ontologically objective general measure of computational intelligence [30], as we want to study its relation with an ontologically objective dichotomous property, which is whether an agent is aware of its phenomenal consciousness. Consequently, any measure that excludes a single property or is noisy or biased, such as the intelligence quotient, cannot be compared with phenomenal consciousness without being also the results biased or noisy. Summarizing the main components of the expression, let $P_{IS,T} + E_{IS,T,C}$ (priors plus experience) represent the total exposure of the system to information about the problem, including the information it starts with at the beginning of training represented by $C$, the curriculum. Let $\omega_T \cdot \theta_T$ be the subjective value we place on achieving sufficient skill at $T$ and let $GD$ be the generalization difficulty for agent *IS* of solving task $T$ given its curriculum or specific properties of the agent $C$:

$$I_{IS,P_{scope}}^{\theta_T} = \mathbb{E}_{P_{scope}}[\omega_T \cdot \theta_T \sum_{C \in Cur_T^{\theta_T}} [P_C \cdot \frac{GD_{IS,T,C}^{\theta_T}}{P_{IS,T}^{\theta_T} + E_{IS,T,C}^{\theta_T}}]]. \tag{3}$$

The formula is basically a generalization of Y that takes into account the previous knowledge, modeled by the curriculum and the priors, to solve a particular task *T*. The difficulty of the task is now modeled by the generalization difficulty and solving a potentially infinite scope is given as the computing the expectation over $P_{scope}$. However, although this measure takes into account whether an entity is able to generalize from prior knowledge as a measure of intelligence, we find the same problems as in the previous measure.

In both measures of intelligence, as the set of potential problems is potentially infinite and not-numerable, any entity would really have a measure of general intelligence of approximately 0, as it would fail to solve a potentially infinite set of problems. Moreover, both measures require having an oracle to determine the difficulty of the task. Consequently, they would both be biased although an objective oracle was able to provide this quantity.

Any measure of intelligence giving any other score rather than zero, although practical, would be just a lower bound of the true intelligence of the entity, better approximated with these measures than with the intelligence quotient measure. Hence, it can be useful for health assessments but never to classify an individual as more intelligent than another individual or, as we will further see, to say that a being is susceptible to having more or less likelihood of having phenomenal consciousness. It is critical to provide an abstract definition of computational intelligence because of two main reasons: First, in order to study its relation with an ontological property such as phenomenal consciousness, it needs to possess the same properties as phenomenal consciousness, that is, being ontological, general and not biased. Second, it can be useful to provide such a definition of intelligence to shed light on the psychology community to provide less biased estimators to it. Once again, this definition corresponds to the parameter, and measures such as the intelligence quotient correspond to the estimator.

### 4.2. Intelligence Quotient and Similar Approaches

Human intelligence includes a series of skills that are focused on different types of problems. The set of problems that human intelligence can solve intersects with the set of computational problems but is not contained in it.

Some examples of this kind of problem include discriminating, which is the most beautiful color for a particular observer in terms of our perception of the colors, which is the best action that we should do in a complex personal conflict involving human relationships, how a person must adapt her emotional state or which is the true notion of a metaphysical phenomenon. The common feature of all these problems is that they involve qualia, information about our universe that Turing machines lack. In particular, we consider qualia as semantic information, in the sense that the observer perceives the quality of color in a particular way, the redness of red, and not in another one. Consequently, this perception can be considered a property that may be codified and that is actually transmitted to the observer by the brain. Although this information is subjective and relative to the observer, it is still information that can be represented in a qualia space such as in the integrated information theory, and is transmitted to the phenomenal consciousness observer.

Consequently, from our point of view, we can only measure the intelligence that a human being shows externally and that is associated with these problems in terms of correlations, which are a reduction of its true scope but are the only way of being objective. Since ancient times, human intelligence has been measured through some of its specific features. For example, in ancient Greece, memory was very valuable. For instance, Plato, in Phaedrus, saw writing as an undesirable tool for external memory, where the memory of dialogues was not only seen as a passive repository of information but also a tool for critical thinking and the creation of new ideas. Then, Rhetoric was indeed a crucial part of education, law, politics, and literature in ancient Rome [50]. In particular, Cicero argued that the ideal orator would be knowledgeable in all areas of human life and understanding, emphasizing the connection between broad knowledge and the ability to speak persuasively, thus highlighting the close relationship between rhetoric and intelligence in Roman society [51]. In the past century, abstract reasoning was

very appreciated and became a critical feature of Stern's intelligence quotient [52]. Stern's intelligence quotient assigns a mental age to a person based on their performance on a series of tests, including reasoning, logic, language, and more. In particular, he divides the scored mental age with the chronological age to obtain a simple ratio.

However, several features that are independent of intelligence may affect Stern's measure. For example, the subject can be in a sad mood, be an introvert or have some special condition such as autism. Due to these conditions, the intelligence shown externally by the subject does not correspond to its true intelligence, in other words, the true human intelligence would be a latent variable contaminated by noise or any approach that measures human intelligence as Stern's intelligence quotient is an approximation to the underlying intelligence of the subject.

Moreover, as Stern's test and similar ones include only a subset of all the subjective problems that a human being is able to solve, the intelligence measured by these tests is a lower bound of the true intelligence of the human being. Consequently, we believe that these approximations are very naive, poor, unreliable, culturally biased, and noisy. From a statistical point of view, the intelligence quotient would be a poor estimator of human intelligence: biased because it does not test all the areas of intelligence and it is influenced by Western culture and with high variance as its measurement contains noise because individuals may be nervous, be shy, have a special condition or simply do not wish to score high.

Hence, as we can only obtain a measure of intelligence via a test analogous to the one of Stern, as the quality of the approximation is poor, the value of this random variable cannot be used in a causal relation with the value of the phenomenal consciousness dichotomous variable. Recall that these tests are only able to reduce the true underlying intelligence of a human being, or even a system, as an approximate lower bound. Consequently, this quantity cannot be established as the cause nor the effect of phenomenal consciousness.

We can illustrate several examples of this statement. For instance, a comatose person is, according to neuroscience, phenomenally conscious [53] but would score a 0 according to Stern's test or similar ones. Another example is natural language generative transformers like GPT-3. This algorithm is very close to passing the Turing test [47] and performs very successfully in intelligence quotient tests. However, as we will see in the next section, it is clear that this system does not possess awareness. Finally, a Down syndrome person would score fewer points on average than a neurotypical person but clearly possesses consciousness. These three examples show how computational intelligence and phenomenal consciousness are not directly related, therefore refuting Russell's analogy. An even more convincing case than the rest is this one: In the science-fiction book *The Three-Body Problem* [54], an enormous plethora of people were displayed on a planet acting like a CPU. Each person acts as a transistor, creating a vast Von Neumann architecture. Most critically, observe that there is no physical connection between the people acting as transistors. Consequently, according to consciousness theories such as information integration theory, which requires physical connections [12], or the Pribram–Bohm holoflux theory of consciousness [55], this people-CPU would be non-conscious as a whole. However, it can solve the same problems that a high-capacity deep learning model can solve, as the people CPU can execute a program that implements the deep learning model. This is the most obvious case where we can see that any algorithm, independent of the degree of intelligence we can measure concerning its behavior, does not have phenomenal consciousness and that phenomenal consciousness does not share a relationship concerning intelligence. In the following section, we will argue how non-computational intelligence may be correlated with consciousness, but it remains a mystery, and we cannot say objectively if they are dependent or not.

## 5. Intelligence Is Not a Measure of Consciousness

If we accept as absurd that they are dependent, we find some problems. We used in all the sections of Bayes theorem to model the two hypotheses, entities having or not having phenomenal consciousness.

*Machine Consciousness*

The computer science community that studies the potential for consciousness in machines is called machine consciousness [20]. In particular, the machine consciousness community, inherits the assumptions of functionalism, like multiple realizability and connectionism to speculate that systems or robots may develop qualia through the implementation of expert systems, machine learning models, hybrid methodologies, or other variants of information processing systems that, in any case, they can be emulated using Turing machines [20]. However, as illustrated in the previous section with the people-CPU, the computational intelligence shown by algorithms, independently of its complexity, is not the cause of phenomenal consciousness. Moreover, as we will illustrate in the following section, there are more philosophical arguments that provide evidence for the highly remote hypothesis that computational intelligence is the cause of phenomenal consciousness.

Our argumentation depends on the assumption that artificial intelligence systems, like high-capacity deep learning models, are not aware of themselves. As currently an ensemble of systems would have greater computational intelligence than human beings and they do not have phenomenal consciousness, this example is a great counter-argument to the hypothesis that phenomenal consciousness is an epiphenomenon of computational intelligence or that they are simply dependent variables. Hence, in this section, we will describe the main counterarguments to the strong artificial intelligence hypothesis, i.e., the one saying that complex machines implementing high-capacity models and reasoning systems may arise consciousness by emergence.

The Nobel laureate Roger Penrose defends its controversial Orch-Or theory that states that phenomenal consciousness arises at the quantum level inside neurons [56]. Roger Penrose argues that human consciousness cannot be replicated by a computer, providing a plethora of strong artificial intelligence counter-arguments that show how understanding and consciousness are non-algorithmic processes and are possibly related to quantum mechanics.

First, we find the famous Searle Chinese room experiment [57], which highlights the difference between pattern recognition, or computation, and understanding. A long list of criticisms has been presented against Searle's original experiment (a search in Google Scholar gives more than 10,000 results). We refer the reader to [58] for a detailed account of responses. Summing up, Searle defends biological naturalism [59], which views consciousness as a biological process and rejects the idea that machines can achieve true consciousness. Also, we would like to focus on Harnad's response [60], which focuses on symbols having a meaning that must be grounded in "robotic functions" that link the system to the world. Later, he argued that even if these robotic functions were implemented, "feeling" would be missing [61].

In our words, understanding a language requires an additional mapping that the observer that lies inside the room lacks. A mapping of every word of the language and the qualia that the words refer to. Qualia are necessary for understanding, and phenomenal consciousness is necessary for qualia. Hence, as the machines lack phenomenal consciousness, they are unable to understand a language and consequently all that they do perform is pattern recognition, in other words, solving complex correlations creating a function whose input is a sentence of a language, and its output is another sentence of that language.

Recall from previous sections, where we provide the example of the people-CPU that appears in the science fiction book *The Three-Body Problem* [54] that was able to perform complex computations and run algorithms to predict a planetary disaster without using computers, that performing complex pattern recognition tasks due to the information processing performed by high capacity deep learning or other statistical models is not

enough to arise phenomenal consciousness by emergence. Concretely, not only in science fiction have we found an example of a person-CPU but in a real experiment available on Youtubeand that has been implemented in a code that is available on Github, we have found how people were organized smartly in a field emulating a brain to perform an algorithmic task. If that experiment had more people available, they could solve any kind of problem that a Turing machine is able to solve. In other words, the Stilwell brain is also a Turing machine, like quantum or classical computers, that does not possess phenomenal consciousness. If an external observer does not know whether Stilwell's brain is a code, like the one hosted on Github, or people being organized in a smart way, it could argue that it is intelligent. However, according also to, without loss of generality, the integrated information theory of consciousness and the Pribram–Bohm holoflux theory of consciousness, the Stilwell brain or any other brain created by independent entities is an excellent example that shows how computational intelligence can not model phenomenal consciousness.

It is also important to consider that all the algorithms that can be executed in a computer can be solved by Turing machines [62]. Quantum computers are not an exception, both classical computers and quantum computers are universal Turing machines and, hence, solve the same kind of problems only with different computational complexity [63]. Nevertheless, humans are able to feel, which requires being able to perceive the qualia of the feeling and, we have said before, having the phenomenal consciousness required to feel, a phenomenon that is not able for a Turing machine and that we cannot measure objectively [64]. If quantum computers are not able to possess the characteristics and abilities of phenomenal consciousness; hence, the idea of the brain being a quantum computer or arising phenomenal consciousness by means of a quantum-like procedure is, at least, a reductionist one, as, in principle, phenomenal consciousness is independent of this procedure.

Finally, because of the qualia that we experience, we can gain intuition about problems that do not have an algorithmic solution. It is especially relevant that this intuition, the experience of being able to understand these problems, cannot be sensed by a computer, as it cannot perceive qualia. Some examples of noncomputable problems are the halting problem (to determine from a random computer program description and an input, whether the program will finish executing the problem, or continue to run forever [65]). Another example is Hilbert's tenth problem, which deals with Diophantine equations (equations involving only sums, products, and powers in which all the constants are integers and the only solutions of interest are integers). In particular, Hilbert's problem, which proved to be undecidable, is described as being able to find an algorithm that decides whether a random Diophantine equation has an integral solution [66].

Dealing with people suffering from different syndromes such as Down [67] or severe autism [68], they would score low in Stern's intelligence quotient and are phenomenally conscious. This is a counter-example to Russell's analogy dealing with computational intelligence. The extreme case would be the one dealing with a comatose person. Concretely, there is empirical evidence coming from neuroscience that shows how comatose people have neural correlates of consciousness, which has been called islands of consciousness [53], conscious states that are neither shaped by sensory input nor able to be expressed by motor output. Technically, people suffering from a comatose state would be phenomenally conscious but unable to perform any kind of movement or reaction to any external stimuli.

Lastly, neurobiology gives us evidence that animal brains share features with our brains dealing with the neural correlates of consciousness [69]. Moreover, neurobiological evidence shows how animal brains perform similarly to us in dealing with the elaboration of the primary emotions, which include the foraging-expectancy system, the anger–rage system, the fear–anxiety system, the separation–distress–panic system, and social–play circuitry [70]. Although they do not solve mathematical problems nor are able to learn human languages, birds have great spatial memory [71], dogs great smell [72], ants great visuals [73]. Hence, computational intelligence cannot model phenomenal consciousness.

### 6. Can Computational Intelligence Model Phenomenal Consciousness?

We will now formalize Russell's analogy from a Bayesian point of view. First of all, we emphasize that we prefer to use the Bayesian framework when we refer to random variables, as phenomenal consciousness. Concretely, from a Bayesian point of view, consciousness is a random variable because we do not observe it directly for other individuals except us; hence, from a Bayesian point of view, we can only reason about it using random variables until it is possible to reduce the uncertainty about this variable via observations, hence being a clear example of a random variable from a Bayesian perspective.

The latent, unobservable measure would be whether an entity possesses phenomenal consciousness or not. We assume here, as we isolate the observer of phenomenal consciousness, in the sense of the defined term awareness by Dehaene, from all the different features of consciousness such as access consciousness, that phenomenal consciousness is a dichotomous variable $C$. Recall that phenomenal consciousness is not being aware of more or fewer phenomena, as the complexity of the integrated information theory qualia space $\phi$ can model. Phenomenal consciousness, from our definition, is being an observer of the qualia space generated by a living being. Consequently, you can only be aware of the qualia space, an observer of the qualia space, or not. Hence, following our assumptions that phenomenal consciousness is not an epiphenomenon or intrinsically related to the qualia space but a property of beings to be aware of their qualia space, we define phenomenal consciousness as a dichotomous variable of perceiving or not the qualia space that a being generates.

Let $I$ be the computational intelligence of an entity as defined in previous sections, denoted by the continuous numerical variable $I$. A subject $S$ may possess or not have phenomenal consciousness, but with the current state of science, we are only able to determine whether it is conscious by looking at the neural correlates of consciousness. If the system does not have a biological brain or nervous system, science is unable to provide any clue about the consciousness of $S$. Then, $p(C \mid S)$ would be the conditional probability that a subject $S$ has phenomenal consciousness such that $p(C = 1 \mid S) + P(C = 0 \mid S) = 1$ and $p(I \mid S)$ is the conditional probability of the computational intelligence of the subject. Concretely, an intelligence quotient test would not determine the intelligence of $S$ as a point estimation but the only thing that it would do is to reduce the entropy of the $p(I \mid S)$ distribution.

In order to carry out this analysis, we use some concepts from probability theory that we now review. The first one is the amount of information needed to encode a probability distribution, also known as entropy. The entropy $H(\cdot)$ can be viewed as a measure of information for a probability distribution $\mathcal{P}$ associated with a random variable $X$. That is, it is self-information. It can be used as a measure of the uncertainty of a random variable $X$. When the random variable is continuous, we refer to the entropy as differential entropy. The entropy of a uni-dimensional continuous random variable $X$ with a probability density function $p(x)$, or differential entropy $H[p(X)]$, is given by the following expression:

$$H[p(X)] = - \int_S p(x) \log p(x) dx. \tag{4}$$

where $S$ is the support of the random variable $X$, that is, the space where $p(x)$ is defined. The entropy $H(\cdot)$ is useful to model the following relation: If we have a random variable $X$ with high entropy $H(\cdot)$, that means that we have low information about the values that it may take. On the other hand, if we consider a random variable $X$ with low entropy $H(\cdot)$, it is a sign that we have high information about the potential values that the variable $X$ can take. In other words, higher knowledge of a random variable implies lower entropy and vice-versa. Another interesting concept regarding information theory, which we use in this work, is the mutual information $I(X; Y)$ of two random variables $X$ and $Y$. Mutual information is defined as the amount of information that a random variable $X$ contains about another random variable $Y$. It is the reduction in the uncertainty of one random variable $X$ due to the knowledge of the other. Mutual information is a symmetric function.

Consider two random variables $X$ and $Y$ with a joint probability density function $p(x, y)$ and marginal probability density functions $p(x)$ and $p(y)$. The mutual information $I(X; Y)$ is the relative entropy between the joint distribution $p(x, y)$ and the marginal distributions $p(x)$ and $p(y)$:

$$I(X; Y) = \sum_x \sum_y p(x, y) \log \frac{p(x, y)}{p(x)p(y)} . \tag{5}$$

Concretely, we define as information gain the amount of information we gain for a particular random variable knowing the value of the other one.

According to Russell, we know that human beings are likely to be conscious, so we denote the being of a human being as the dichotomous random variable $B$. Then, $p(C = 1 \mid B = 1) = 1$ independently on the degree of intelligence. More technically, the information gain of the intelligence degree $I$ over consciousness, given that the entity is a human being, is 0.

$$IG(C, I \mid B = 1) = 0. \tag{6}$$

In other words, the entropy $H(\cdot)$ of the conditional probability distribution of consciousness is also conditioned to the degree of computational intelligence of subject $S$, which is also a random variable as we do not have direct access to it, is the same one. Then, in our case, we can illustrate that the entropy on the consciousness random variable for humans $H(C \mid B = 1)$ is equal to the conditional entropy on the consciousness for a certain computational intelligence level $I$.

$$H(C \mid B = 1, I) = H(C \mid B = 1). \tag{7}$$

As $p(C = 1 \mid B = 1) = 1$, there is no need to show that $H(I \mid B = 1, C = 1) = H(I \mid B = 1)$, as it is obvious. Hence, we have formally shown how, in the case of human beings, the computational intelligence degree is independent of the phenomenal consciousness variable. However, until now we have only performed the analysis of computational intelligence and phenomenal consciousness in the case that the subject is a human being. Nevertheless, important implications of this analysis need to be taken into account. For example, we now know that a low measure of computational intelligence, according to the intelligence quotient of Stern, does not condition the subject from being conscious. Let $p(I) <<$ denote a probability distribution over the computational intelligence for a subject $S$ having its density concentrated over a low value. Concretely, we know that $p(C = 1 \mid p(I) <<) = 1$. We put here $p(I)$ and not $I = k$ being $k$ a real number as we have denoted that current measures of intelligence are noisy lower bound over the true value of intelligence of subject $S$, which is a random variable. Importantly, we now know with complete certainty that, in the case of disabilities or certain comatose states, a subject has phenomenal consciousness.

Next, we analyze and compare the probability distributions $p(C \mid I)$ and $p(C)$. Science gives us evidence that if the entity shares features with the human being biologically speaking, concretely the neural correlates of consciousness, the subject may be conscious. We denote with $N \in [0, 1]$ a continuous numerical variable that represents the degree of biological similarity of the brain of the subject with the brain of the human being. Concretely, current AI systems, denoted with the dichotomous variable $A$, have $A = 0$, as deep neural networks or meta-learning methodologies are just sequences of instructions sequentially computable by Turing machines as we have shown before, although their name may be misleading. In particular, every algorithm written in a computer can be solved by a Turing machine. Critically, the concept of the Turing machine is relevant to the question of machine consciousness because it provides a framework for thinking about the limits of computation and because it models correctly any algorithm performed in a computer. One way in which the Turing machine can be used to clarify the question of machine consciousness is by providing a way to distinguish between computational processes that are purely mechanical, such as the ones involved in intelligence as neuroscience shows [74], and those that involve some form of conscious, or meta-cognition, experience, whose

physical explanation is not clear [64]. An infinite tape quantum Turing machine would be able to run a set of algorithms that potentially solve any problem related to intelligence but whose artificial support would not necessarily perceive qualia, always remaining unaware of solving any problem. Also, according to some theories of consciousness, such as the global workspace theory, consciousness involves the integration of information from different parts of the brain. From this perspective, a machine could be considered conscious if it is capable of integrating information in a similar way. However, it is not clear whether a Turing machine, which eventually operates on a fixed set of rules, is capable of this type of integration. Further work will explain other modeling alternatives like Gualtiero Piccinini, Nir Fresco, and Marcin Milkowski.

We found a real analogy with $P(C)$ and $P(C \mid N)$. Concretely, these variables are, according to evidence found in neurobiology, linearly correlated, i.e., $r(C, N) \approx 1$ being $r$ the correlation coefficient. However, a bird, elephant, dolphin, monkey, or cephalopod, for example, may score a low computational value $p(I) <<$. However, again, we find that conditioning the variable $p(I) <<$ to the conditional distribution $P(C \mid N)$ does not change the entropy of the distribution:

$$H(C \mid N, p(I) <<) = H(C \mid N). \tag{8}$$

Finally, we use the example of a meta-learning system to show how the degree of computational intelligence is not correlated to phenomenal consciousness. Concretely, a meta-learning system with $N = 0$ has the biggest computational intelligence known as it is able to solve a potentially infinite set of computational problems that humans or animals are not able to solve up-to-date, as we have seen in previous sections. We denote that such a system has a computational intelligence probability distribution $p(I) >>$. However, we know that $p(C = 0 \mid N = 0) = 1$, independently of its degree of computational intelligence. In other words, if we condition the probability to $p(I) >>$, for all the set of artificial intelligence systems, we have that $p(C = 0 \mid N = 0) = p(C = 0 \mid N = 0, p(I) >>) = 1$. Hence, the degree of intelligence does not generate phenomenal consciousness as an epiphenomenon or by emergence. Concretely, it is the anatomy of the biological brain, or even less probably the nervous system or body, where supposedly we find, at least, neural correlates of consciousness. Given all the information and evidence that we have provided, we could formalize that the information gain of the computational intelligence random variable given that we know the phenomenal consciousness variable if we marginalize the kind of entity that may have phenomenal consciousness is 0, i.e., they are independent random variables independently of the degree of intelligence.

$$IG(C, I) = 0. \tag{9}$$

From a Bayesian point of view, this information could be formalized as follows. Concerning artificial intelligence systems, let $p(C = 1 \mid I, N = 0)$ be an a priori distribution representing the probability of the system being conscious, our previous beliefs coming from Russell's analogy. Following this analogy, this probability was high as the system is intelligent and the complementary probability, $p(C = 0 \mid I, N = 0)$, is low. We have provided empirical and theoretical evidence showing that this is not true, which we formalize in the likelihood $p(E \mid C = 1, I, N = 0)$, being $E$ the evidence we have illustrated in previous sections. Let $p(E)$ be the marginal likelihood representing the probability of our evidence being true, which is high due to the fact that it comes from highly cited papers of various research communities like neurobiology, psychiatry, or philosophy of mind. Lastly, let $p(C = 1/E, I, N = 0)$ be our posterior beliefs of the hypothesis that artificial intelligence systems are conscious. As the probability rectifier coefficient is very low, that is $p(E \mid C = 1, I, N = 0)/p(E)$, despite having an a priori belief supporting the hypothesis of conscious artificial intelligence systems, now the posterior belief clearly shows that $p(C = 0/E, I, N = 0) > p(C = 1/E, I, N = 0)$ significantly. Mainly because computational intelligence cannot model phenomenal consciousness.

We would like to discuss several non-conventional computing paradigms under this framework. The first one is symbolic computation. In particular, recall that first-order logic framework is a subset of Bayesian inference, as for example, we have $p(B|A) = 1$ is equal to $A \rightarrow B$. However, without having to use fuzzy logic, we could have a Bayesian network $p(B|A) = 0.5$, being impossible to model under classical first-order logic. As probabilistic programming is also a subset of the algorithms that can be run in a Turing machine, all the previous statements hold. Second, current deep learning models and spiking neural networks, although being fancy for folklore psychology and with brilliant behavior, are also reducible to binary instructions being executed in a computer, very different from how a human brain works, so they also belong to Turing's machine algorithms and previous statements hold. Finally, we make an exception for future neuromorphic hardware, as the span of algorithms that may execute and the physical and chemical properties involved may, we do not know, emerge consciousness. Hence, we place a non-informative prior to the emergence of consciousness in future neuromorphic hardware systems, leaving the analysis of these systems for further research.

### 7. Conclusions and Further Work

Phenomenal consciousness is defined as the awareness of an individual of internal and external stimuli, of the information processed by the brain, in the form of qualia. In this work, we have analyzed and shown how Russell's analogy of consciousness, which basically states that awareness and intelligence are correlated with high probability, is a fallacy, at least for computational intelligence. In order to do so, first, we defined what is phenomenal consciousness and give an objective measure of computational intelligence. Then, we provided a set of counter-arguments to Russell's analogy with evidence coming from neurobiology, psychiatry, or philosophy of mind, where we can see how computational intelligence cannot model phenomenal consciousness. Consequently, we include a formalism with probability and information theory to represent this relation. We wonder, for future work, about the implications of these statements regarding society and ethics. Another research line is a formal comparison of this work against some consciousness theories such as Integrated Information Theory.

**Author Contributions:** Conceptualization, E.C.G.M. and S.L.; Methodology, S.L.; Investigation, E.C.G.M. and S.L.; Resources, S.L. All authors have read and agreed to the published version of the manuscript.

**Funding:** This research received no external funding.

**Institutional Review Board Statement:** Not applicable.

**Informed Consent Statement:** Not applicable.

**Data Availability Statement:** Not applicable.

**Conflicts of Interest:** The authors declare no conflict of interest.

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
