# Peer review of "Can Computational Intelligence Model Phenomenal Consciousness?"

_philosophies, doi:10.3390/philosophies8040070_

Round 1

Reviewer 1 Report

This is an interesting point of view on a difficult topic. However, there are issues that would improve the impact of the paper

1) The word 'independent' appears in the title and throughout the paper. What is really meant is "Can computational intelligence model phenomenal consciousness". Do consider making changes to clarify your work with an appropriate title. 

2) The paper rests on the lack of validity of current work on machine consciousness.  However the references given for this are narrow and should be broadened and assessed as to whether they alter the argument of the paper. Including the following might help

Neural Netw.  2013 Aug;44:112-31.

 doi: 10.1016/j.neunet.2013.03.011. Epub 2013 Mar 26.

The rise of machine consciousness: studying consciousness with computational models

James A Reggia 

3) The paper refers to computational intelligence being Turing related algorithmic computation. However, to approach Consciousness it has been argued that this is an emergent property of dynamic neural networks. Please consider reviewing

Aleksander and Morton 

Phenomenology and digital neural architectures

Neural Networks

Volume 20, Issue 9, November 2007, Pages 932-937

4) The authors have stated that some mathematical expressions are included because this is required by modern paper writing.   Please exclude such a comment amd make sure that the expressions are necessary to the argument. This is not evident.

A read-over by a native English speaker would be helpful

Author Response

Thanks for your useful suggestions. We have considered them all for the paper and modify it according to your observations. In particular, regarding “The word 'independent' appears in the title and throughout the paper.” we have changed more than 20 statements related to independence, changing them to relation or model. Also, having in mind the observations of another reviewer, our idea with the paper is to provide theoretical and empirical of the idea that phenomenal consciousness is independent from computational intelligence, or at least relaxing this statement as we have done in this second version of the paper, that phenomenal consciousness cannot be modelled by computational intelligence. In particular, we have stated in the 4th section a definition of computational intelligence, not just intelligence in general which is a concept so wide and general that different communities do not share the same concept of it. Under our specific assumptions (we talk about phenomenal, not just consciousness, or computational, not just intelligence) we find that our paper provides a set of evidence and use it from the point of view of the Bayesian framework that justifies its novelty.

We have rewritten the introduction section and clarified the paper in the respects pointed by the reviewer with respect to the machine consciousness literature. We thank his/her comment. We have introduced this perspective and included the paper mentioned by the reviewer, which was very interesting indeed for our research.

Concerning the fact that we have stated that some mathematical expressions are included because this is required by modern paper writing, we have removed the statement from the paper. The introduction has hopefully clarified the taxonomies and the whole paper has been reviewed for better English and readability.

Reviewer 2 Report

While the article starts from a sound point of view - that consciousness and intelligence are not interdependent - and it is a fine and well-researched one, I detected a few problems that make the argument a bit difficult to follow. I am listing them below with the hope that I can be of help and not to criticise. I myself have the same problems - especially with the first drafts of my articles - then things improve :)

Here are the problems:

- too wide generalisations and even small contradictions or things that are already clear presented as something newly discovered/ demonstrated ... for instance: - consciousness and intelligence presented as dependent according to 'folk psychology', but also to the 'psychology community', 'philosophy of mind community', 'computer science community' in general ... is that really so?

- phenomenal consciousness as independent from computational intelligence - I thought that was clear - especially if we look at the entire 'embodied cognition' trend of research, stating that even cognition is not generated only by intelligence or, specifically, by mental representations - but the author presents the idea as some new and original idea of his/ her own, if I understand him/ her well - I think he/ she should place it in a bit of a context

- a bit of an awkward taxonomy - lines 28-37

- par. 3 - on Russell's analogy of consciousness - it is not clear from the quotation and the explanations of the author that Russell really said that ... and there are contradictions or fallacious arguments regarding this - for instance, behaviour is a cause of consciousness and bahaviour is also a cause of intelligence and from that can be inferred, according to the author, that intelligence is an effect of consciousness (see lines 401-402 - maybe I am wrong ...) while later he/ she states that consciousness may be considered an effect of intelligence, but he/ she actually challenges that; things need to be explained with more clarity anyway - later on the author pleads for 'correlation' and not for 'causality' - but it is not very clear for the reader which is the author's p.o.w. and which is his/ her sources' p.o.w.

- 'intelligence' - defined too late - at par. 4

- par. 4 - again generalizations - memory, rhetoric, intelligence - 372-375

- 454-472 - jumping from Penrose 1991 to Searle 2009 (nn. 49-50) without clearly explaining which says what and which is the author's view regarding their p.o.w.s

- a lot of demonstration (Bayesian?) is then based, in par. 6, on these very general and not clearly specified taxonomies - it would be good to have them clarified also for the sake of this demonstration being better understood

I think that English and argumentation should be revised a bit for the sake of a better presentation of this fine article.

Good luck and all the best!

A few infelicities of English - for instance:

- 'phenomenons' instead of 'phenomena' (27)

- subject-predicate agreement problems - for instance: 36-37 (phenomena ... it is related) and also 273-275...

- see also 81-84 for phrases that are not very clear

- 'has sense' instead of 'makes sense', 'a epistemology ...' instead of 'an epistemology' ...

- 'independence with' instead of 'independence from'

- etc.

Round 2

Reviewer 1 Report

Thank you for adaptingyour paper according to suggestions.